# MK2 Inhibition Induces p53-Dependent Senescence in Glioblastoma Cells

**DOI:** 10.3390/cancers12030654

**Published:** 2020-03-11

**Authors:** Athena F. Phoa, Ariadna Recasens, Fadi M. S. Gurgis, Tara A. Betts, Sharleen V. Menezes, Diep Chau, Kristiina Nordfors, Joonas Haapasalo, Hannu Haapasalo, Terrance G. Johns, Brett W. Stringer, Bryan W. Day, Michael E. Buckland, Najoua Lalaoui, Lenka Munoz

**Affiliations:** 1School of Medical Sciences, Charles Perkins Centre and Faculty of Medicine and Health, The University of Sydney, Sydney, New South Wales 2006, Australia; apho6944@uni.sydney.edu.au (A.F.P.); ariadna.recasens@sydney.edu.au (A.R.); fadimaged@hotmail.com (F.M.S.G.); taraabetts@gmail.com (T.A.B.); Sharleen.menezes@sydney.edu.au (S.V.M.); Michael.buckland@sydney.edu.au (M.E.B.); 2Inflammation Division, The Walter and Eliza Hall Institute of Medical Research, Parkville, Victoria 3052, Australia; dieptchau@gmail.com (D.C.); lalaoui@wehi.edu.au (N.L.); 3Department of Medical Biology, University of Melbourne, Parkville, Victoria 3050, Australia; 4Department of Pediatrics, Tampere University Hospital, 33521 Tampere, Finland; kristiina.nordfors@gmail.com; 5Tampere Center for Child Health Research, University of Tampere, 33014 Tampere, Finland; 6The Arthur and Sonia Labatt Brain Tumour Research Centre, The Hospital for Sick Children, Toronto, ON M5G 0A4, Canada; joonas.haapasalo@gmail.com; 7Department of Pathology, Fimlab Laboratories, Tampere University Hospital, FI-33521 Tampere, Finland; hannu.haapasalo@fimlab.fi; 8Oncogenic Signalling Laboratory, Telethon Kids Institute, Perth Children’s Hospital, 15 Hospital Avenue, Nedlands, WA 6009, Australia; terrance.johns@telethonkids.org.au; 9QIMR Berghofer Medical Research Institute, 300 Herston Road, Herston, QLD 4006, Australia; brett.w.stringer@gmail.com (B.W.S.); bryan.day@qimrberghofer.edu.au (B.W.D.); 10Brain and Mind Research Institute, The University of Sydney, Sydney, New South Wales 2006, Australia

**Keywords:** glioblastoma, MK2, p53, senescence, temozolomide

## Abstract

MAPK-activated protein kinase 2 (MK2) has diverse roles in cancer. In response to chemotherapy, MK2 inhibition is synthetically lethal to p53-deficiency. While *TP53* deletion is rare in glioblastomas, these tumors often carry *TP53* mutations. Here, we show that MK2 inhibition strongly attenuated glioblastoma cell proliferation through p53^wt^ stabilization and senescence. The senescence-inducing efficacy of MK2 inhibition was particularly strong when cells were co-treated with the standard-of-care temozolomide. However, MK2 inhibition also increased the stability of p53 mutants and enhanced the proliferation of p53-mutant stem cells. These observations reveal that in response to DNA damaging chemotherapy, targeting MK2 in p53-mutated cells produces a phenotype that is distinct from the p53-deficient phenotype. Thus, MK2 represents a novel drug target in 70% glioblastomas harboring intact *TP53* gene. However, targeting MK2 in tumors with *TP53* mutations may accelerate disease progression. These findings are highly relevant since *TP53* mutations occur in over 50% of all cancers.

## 1. Introduction

MAPK-activated protein kinase 2 (MAPKAPK2 or MK2) is activated via p38 MAPK-mediated phosphorylation in response to stress and exhibits diverse roles in inflammation and cancer [1]. Pro-tumorigenic MK2 functions include the deregulation of apoptosis in skin tumors [2], enhanced metastasis of breast tumors [3] and polarization of macrophages into their M2 state that promotes angiogenesis in colorectal cancer [4]. MK2 has been also successfully targeted to improve the efficacy of radiation therapy [5] and various cancer drugs, such as cisplatin [6], doxorubicin [7], Chk1 kinase inhibitors [8] and Smac mimetics [9].

Although a number of different mechanisms by which MK2 promotes tumor growth and therapy resistance have been reported, there are limited data on MK2 signaling in gliomas [10,11,12,13,14]. Gliomas are classified in a range from pilocytic astrocytoma (Grade I), to diffusely infiltrating lower-grade gliomas (Grades II-III) and to high-grade glioblastoma (Grade IV) [15]. Although patients with lower-grade gliomas have better survival prognosis than glioblastoma patients, lower-grade glioma is a uniformly fatal disease in young adults, as all tumors eventually progress to secondary glioblastomas and cause death within 7 years [16]. The median survival of patients with primary glioblastoma remains 15 months [17]. Thus, there is a critical need for new therapeutic targets.

We have shown that active MK2 is expressed in clinical glioblastoma specimens and that p38 -MK2 regulates inflammation in the presence of the EGFRvIII and HRasG12V mutants [10,11]. In glioma xenografts, anti-tumour activity of p38 MAPK inhibitor LY2228820 was linked to the reduction of inflammatory tumour microenvironment [12]. However, it is unknown whether MK2 has other non-inflammatory roles in glioblastoma and whether tumors depend on MK2 signaling in response to temozolomide, which remains the mainstay of glioblastoma treatment. Temozolomide is a DNA-alkylating agent that causes persistent DNA strand breaks and replication fork collapse. While temozolomide efficacy is restricted to the subset of patients with silencing of the DNA repair protein methyl-guanine-methyl-transferase (MGMT) [18], numerous MGMT-independent mechanisms of temozolomide resistance have been reported [19,20,21,22].

In response to DNA-damaging therapy, MK2 inhibition is synthetically lethal to p53-deficiency. In the absence of functional p53, the MK2 pathway is essential to halt cell division, enabling DNA repair and allowing cancer cells to proliferate. Thus, MK2 inhibition in cells lacking p53 caused the bypass of cell cycle arrest and the accumulation of unrepaired DNA damage, thereby increasing the cytotoxicity of chemotherapy [7,23,24]. These studies also suggested that MK2 inhibition could be synthetically lethal to tumors with mutated p53; however, this remains unsubstantiated. While *TP53* deletion is rare in gliomas (<1%), p53 mutations occur in 48–91% of lower-grade gliomas and ~30% of glioblastomas [25,26]. To investigate whether MK2 inhibition is synthetically lethal to p53 mutations, we sought to delineate the MK2 signaling in p53^wt^ and p53-mutated glioblastoma cells. Our work shows that glioma patients with the highest MK2 activity had the worst survival rate and identifies the role of MK2 in glioblastoma cell proliferation and in response to the standard-of-care, temozolomide.

## 2. Results

### 2.1. MK2 Activity Correlates with Poor Glioma Prognosis

To investigate the relevance of MK2 in gliomas, we examined the Cancer Genome Atlas (TCGA) and Genotype-Tissue Expression (GTEx) datasets using Gene Expression Profiling Interactive Analysis (GEPIA) [27]. MK2 mRNA expression was significantly higher in lower-grade gliomas (LGG) and glioblastomas (GBM) compared to the expression in normal brain tissue (Figure 1A and Appendix A). An Oncomine analysis of the TCGA [25] and Sun brain [28] databases confirmed over-expression of the MK2 gene in glioblastoma (Appendix A). Furthermore, patients with high MK2 mRNA expression (top 25%) exhibited shorter disease-free (Figure 1B,C) and overall (Figure 1D,E) survival.

We next analyzed the MK2 expression and activation in 126 tissue samples from 60 glioma patients. At the protein level, 125/126 (99%) samples showed positive MK2 immunostaining (Figure 1F). Highest MK2 expression (score 3+) was observed in 27% of Grade I, 39% of Grade II, 47% of Grade III and 39% of Grade IV tumors (Figure 1G). These 3+ tumors were considered as ‘positive’, and the rest were grouped as ‘negative’ in the correlation analyses. While the level of MK2 expression did not correlate with the tumor grade, patient age or gender, MK2 was strongly expressed in secondary (*p* = 0.009; chi-square test) and IDH1-positive (*p* = 0.013; chi-square test) glioblastomas (Appendix A). In parallel, we found that 87/118 (74%) samples showed positive staining for active phospho-Thr334 MK2 (p-MK2) (Figure 1F). When analyzing the moderate (2+) and strong (3+) p-MK2 scores, 27% of Grade I, 24% of Grade II, 47% of Grade III and 37% of Grade IV were expressing p-MK2 (Figure 1G). As seen in the MK2 analysis, the p-MK2 expression did not correlate with the tumor grade, patient age or gender (Appendix A). However, the p-MK2 expression correlated with recurrent glioblastomas (*p* = 0.049, chi-square test; Appendix A).

For the survival analysis, the cohort was divided into high MK2/p-MK2 (score 3+) groups and low MK2/p-MK2 (score 0, 1+, 2+) groups. Although patients with high MK2 mRNA levels had shorter survival times (Figure 1B–E), the MK2 protein did not correlate with the patient survival (*p* = 0.081, log-rank test). However, patients with high expression of p-MK2 had the worst survival (*p* = 0.027, log-rank test, Figure 1H). In summary, MK2 is over-expressed in gliomas and high MK2 activity correlates with the poor prognosis of glioma patients.

### 2.2. MK2 Inhibition Attenuates the Proliferation of p53^wt^ Glioblastoma Cells

To investigate the role of MK2 in glioblastoma cell proliferation and in response to temozolomide, we first confirmed that temozolomide activates MK2 (p-MK2, Figure 2A) and MK2 down-stream target Hsp27 (p-Hsp27, Figure 2B). We then depleted MK2 in p53^wt^ A172 cells (Figure 2B) and in U251 cells harboring a gain-of-function mutant p53^R273H^ (Figure 2C). Decreased phosphorylation of the MK2 downstream target Hsp27 confirmed transfection efficacy. We found that MK2 knockdown in p53^wt^ A172 cells significantly reduced clonogenic survival in the absence or presence of temozolomide (Figure 2D). In contrast, MK2 knockdown in the p53^R273H^ U251 cells did not affect the clonogenic survival and did not improve efficacy of temozolomide (Figure 2E). Following this initial set of experiments, we used temozolomide up to 50 μM concentration, which corresponds to the maximal temozolomide concentration found in human brain tumors following a clinical dosing regimen of 75–200 mg/m^2^ [29,30].

To recapitulate the effectiveness of MK2 inhibition in A172 cells, we employed an ATP-competitive MK2 inhibitor PF-3644022 [31] and an allosteric MK2 inhibitor IV (MK2i) [32]. Both inhibitors attenuated Hsp27 phosphorylation by >50% (Figure 2F). As single agents, only MK2i significantly decreased the clonogenic survival of p53^wt^ A172 cells (Figure 2G). Importantly, when combined with 5 μM of temozolomide, both MK2 inhibitors improved temozolomide efficacy (Figure 2G). Conversely, both MK2 inhibitors failed to exhibit any effect on the clonogenic survival and temozolomide efficacy in p53^R273H^ U251 cells (Figure 2H).

To further validate that MK2 inhibition attenuates the proliferation of p53^wt^ cells, we employed CRISPR/Cas9 gene editing technology to knock out MK2 in cells (Appendix A). Two p53^R273H^ U251 clones (A and B) were completely lacking MK2 (MK2^null^) and showed a reduction of Hsp27 phosphorylation (Figure 2I). However, although we successfully infected p53^wt^ A172 cells with two different MK2 sgRNAs (Appendix A), we failed to generate MK2^null^ clones (Appendix A). We, therefore, infected p53^wt^ U87 cells and selected two clones (A and C) that lacked MK2 expression and showed decreased Hsp27 phosphorylation (Figure 2J). Decreased proliferation of p53^wt^ MK2^null^ U87 clones was confirmed by Ki67 staining (Figure 2K) and in the spheroid growth assay (Figure 2L). As expected, the loss of MK2 failed to affect the growth of the p53^R273H^ U251 spheroids (Figure 2M). These results suggest that MK2 inhibition attenuates the proliferation of glioblastoma cells harboring p53^wt^. This effect was observed in undamaged and in DNA-damaged (i.e., temozolomide-treated) glioblastoma cells.

### 2.3. MK2 Inhibition Induces Senescence in p53^wt^ Glioblastoma Cells

To understand the mechanism underlying the cellular response to MK2 inhibition, we first questioned whether MK2 inhibition contributes to the accumulation of DNA damage, as has been observed in other studies [6,23,24]. Surprisingly, temozolomide-treated p53^wt^ A172 cells showed robust γ-H2AX foci confirming DNA damage; however, this was decreased by silencing MK2 (Appendix A). Similarly, MK2 depletion failed to promote mitotic arrest (Appendix A), another feature associated with the accumulation of DNA damage. These findings indicated that the antiproliferative efficacy of MK2 inhibition is not caused by the accumulation of DNA damage.

Given that the p53^wt^-mediated response includes cell cycle arrest, apoptosis and senescence [33], we investigated these phenotypes upon MK2 inhibition. Cell cycle analysis revealed that MK2 knockdown or inhibition significantly increased the percentage of p53^wt^ A172 cells in the G_0_/G_1_ phases (Figure 3A,B). Increased sub-G_1_ and G_0_/G_1_ fractions were also observed in temozolomide-treated p53^wt^ A172 cells when MK2 was inhibited (Figure 3A,B). However, MK2-depleted p53^R273H^ U251 cells exhibited cell cycles that were identical to the scramble-transfected cells (Figure 3C). While MK2 inhibition alone did not result in apoptosis in p53^wt^ A172 cells; MK2 inhibition increased temozolomide-induced apoptosis (Figure 3D,E). This is in line with the flow cytometry data showing increased sub-G_1_ fraction in co-treated cells (Figure 3A). An analysis of the senescence-associated β-galactosidase (β-gal) and histone 3 lysine 9 trimethylation (H3K9me3) mark [34] revealed that MK2 inhibition induced senescence in ~20% of p53^wt^ glioblastoma cells (based on β-gal quantification) and this increased to ~50% upon 10 days temozolomide treatment (Figure 3F–H). The percentage of β-gal positive senescent cells further increased to ~80% when p53^wt^ MK2^null^ U87 cells were treated with temozolomide for 20 days (Figure 2I). These data suggest that the proliferation-suppressive effect of MK2 inhibition in p53^wt^ cells was predominantly due to the activation of senescence. However, we could not detect senescent cells in p53^R273H^ MK2^null^ U251 clones (Appendix A).

Given the synthetic lethality between MK2 and p53 deletion in non-small-cell lung cancer models [6,7], we queried whether the lack of response of p53^R273H^ glioblastoma cells is due to the different histological cancer type (lung vs. brain) or different p53 status (mutation vs. deletion). To this end, we depleted p53^wt^ in parental and MK2^null^ U87 glioblastoma cells. In line with its tumor-suppressive role, p53 deletion increased the clonogenicity of parental U87 cells but weakly improved the efficacy of temozolomide (Figure 3J, brown bars). Temozolomide-treated p53^wt^ MK2^null^ U87 cells demonstrated decreased clonogenicity (Figure 3K, black bars), which agrees with onset of senescence of temozolomide-treated MK2^null^ U87 clones (Figure 3I). As in parental MK2-expressing cells, knockdown of p53^wt^ in MK2^null^ cells increased clonogenic survival. Importantly, knockdown of p53^wt^ in MK2^null^ clones rendered U87 glioblastoma cells significantly more sensitive to temozolomide (Figure 3K, brown bars). To probe the mechanism of differential chemosensitivity, i.e., MK2 deficiency induces senescence (Figure 3G–I), MK2 + p53 co-deficiency is reported to induce apoptosis [6]**,** we monitored the extent of apoptosis by measuring cleaved PARP (c-PARP). As expected, only U87 cells lacking both MK2+p53 and treated with temozolomide displayed c-PARP (Figure 3L,M). These glioblastoma data confirm synthetic lethality (i.e., apoptosis) between MK2 and p53 deletion, but also implicate that p53 mutation and p53 deletion evoke different cellular responses to MK2 inhibition.

### 2.4. MK2 Inhibition Increases the Expression of Wild-Type and Mutated p53

Since MK2 activates p53-degrading HDM2 ligase through the phosphorylation of Ser166 [35], we examined the effect of MK2 inhibition on the HDM2-p53 pathway. Inhibition of MK2 reduced HDM2 phosphorylation, which was accompanied with increased expression of both wild-type and mutated p53 (Figure 4A–D). Furthermore, loss of MK2 increased the stability of both p53^wt^ and p53^R273H^ when compared to parental cells expressing MK2 (Figure 4E). These data indicate that MK2 inhibition results in HDM2 deactivation and p53 stabilization. Importantly, this mechanism is shared by both p53^wt^ and p53^R273H^ mutants.

Marked colocalization of the p53^wt^ and DAPI signals in A172 cells suggested that the increase of p53 levels occurs mainly in the nucleus (Figure 4F). To confirm, we examined the nuclear accumulation and phosphorylation of p53 at Ser15, which is required for p53-transcriptional activity [36]. p53^wt^ A172 cells treated with two orthogonal MK2 inhibitors displayed a marked p-p53^wt^ increase in the nuclear fractions (Figure 4G). Similarly, we observed an increase in nuclear p-p53^wt^ in the MK2^null^ U87 clones compared to that in parental MK2-expressing cells (Figure 4H). We also assessed the effect of MK2 inhibition on the transcription of p53 target genes that control senescence (CDKN1A, PML, YPEL3) and apoptosis (BBC3, BAX, PMAIP1) [37]. Except for BAX, loss of MK2 increased transcription of p53 target genes with or without temozolomide treatment (Figure 5A–D). In line with increased mRNA expression, MK2 knockout in p53^wt^ cells resulted in increased levels of p21 (coded by CDKN1A), PUMA (coded by BBC3) and NOXA (coded by PMAIP1), while BAX levels remained unchanged (Figure 5E).

### 2.5. p53^wt^ Silencing Rescues the Antiproliferative Effect of MK2 Inhibition

The data in Figure 2, Figure 3 and Figure 4 suggest that p53^wt^ is crucial for MK2 function in glioblastoma. To ascertain that MK2 inhibition attenuates cell proliferation via p53^wt^ signaling, we depleted p53^wt^ in parental and MK2^null^ U87 cells. Importantly, while p21 and PUMA expression increased in MK2^null^ cells, this effect was lost in MK2^null^ cells lacking p53^wt^ (Figure 5F). Furthermore, the decreased proliferation in MK2^null^ U87 clones was also lost upon the deletion of p53^wt^ (Figure 5G), in line with increased clonogenic survival of p53-depleted MK2^null^ cells (Figure 3K). These results suggest that p53^wt^ is critical for the upregulation of tumor-suppressive proteins and the anti-proliferative response to MK2 inhibition. This is supported by the analysis of p53^R273H^ MK2^null^ U251 clones, which revealed that the increased p53^R273H^ expression in the nucleus (Figure 5H) was not accompanied by increased p21 levels (Figure 5I) or senescence (Appendix A).

### 2.6. MK2 Inhibitors Induce Senescence in p53^wt^ Glioblastoma Stem-Like Cells

The data derived from A172, U87 and U251 cells demonstrate that MK2 inhibition stabilizes both wild-type and mutated p53 (Figure 4). In line with p53^wt^ tumor-suppressive functions, MK2 inhibition in p53^wt^ cells induced apoptosis (in up to 30% cells) and senescence (in ~80% cells; Figure 3). However, the stabilization of p53 mutants (Figure 4E) is necessary for their oncogenic activities [38]. As such, MK2 inhibition might result in two opposite effects, depending on the status of the *TP53* gene.

To examine the intact TP53 as a biomarker for the clinical use of MK2 inhibitors, we employed serum-free patient-derived glioblastoma stem-like cells [39,40]. RN1, HW1 and RKI1 cells harbor an intact *TP53* gene; however, they carry a homozygous *CDKN2A* deletion, thereby not expressing the p14/ARF protein, which promotes HDM2 degradation and p53 stability [41]. As such, these cells endogenously express low p53^wt^ levels (Figure 6A). Inhibition of MK2 in p53^wt^ RN1 and HW1 cells induced a minor increase in the p53 expression in the absence of temozolomide, but increased p53^wt^ expression in temozolomide-treated cells (Figure 6A–C). In functional assays, MK2 inhibitors reduced clonogenic survival and induced senescence in p53^wt^ cells, with the most prominent effect observed in temozolomide co-treated cells (Figure 6D–G). Consistent with previous findings in p53^R273H^ U251 cells, MK2 inhibition in p53^R110L^ JK2 stem-like cells reduced the phosphorylation of HDM2 leading to an increase of p53^R110L^ expression (Figure 6H). In line with the p53^R110L^ gain-of-function [33], JK2 spheroids displayed 10% increased growth when treated with MK2 inhibitor, even in the presence of temozolomide (Figure 6I). Taken together, MK2 inhibitors reduced the clonogenic growth of glioblastoma cells harboring the intact *TP53* gene.

Finally, we examined the MK2-p53 relationships in TCGA datasets using cBioPortal for Cancer Genomics platform. As MK2 function depends on the level of the active phosphorylated protein which controls p53 expression via protein stabilization (Figure 4), we questioned correlations at the protein levels. Protein expression for MK2 is not available in these datasets; however, they contained the protein profiles of p53 and p38α MAPK, the only upstream MK2-activating kinase identified to date [42]. The total p38α MAPK (MAPK14) expression did not correlate with p53 expression (Appendix A). Importantly, the expression of phosphorylated p38α MAPK negatively correlated (*p* < 0.005) with p53 expression in both lower-grade glioma and glioblastoma datasets (Figure 6J,K). These negative correlations indicate that active p38α MAPK, which is the unique activator of MK2, leads to the decreased p53 expression, which is in line with mechanistic data presented here. 

## 3. Discussion

MK2 is increasingly appreciated as a non-oncogenic kinase that contributes to different malignant processes in various cancers. Herein, we show that MK2 plays a role in degrading the p53 protein, the main determinant of cell fate. Importantly, we provide evidence suggesting that the MK2-dependent degradation of p53 appears to be a general regulatory mechanism for both wild-type and mutated p53. While wild-type p53 is a potent tumor suppressor, mutated p53 proteins exhibit gain-of-function oncogenic properties.

In p53^wt^ cells, MK2 inhibition attenuated glioblastoma cell proliferation. Mechanistically, this effect was achieved through deactivating HDM2 ligase, leading to increased p53^wt^ expression; this promoted the transcription of tumor-suppressive genes. In untreated cells with low levels of DNA damage, the inhibition of MK2 resulted in senescence, whereas in the presence of externally induced DNA damage, the cells responded to MK2 inhibition by activating both apoptotic and senescent programs. The negative regulation of p53^wt^ by MK2 described herein provides a molecular mechanism for the pro-tumorigenic functions of MK2 in p53^wt^ glioblastomas, extending the previous data showing tumor progression through MK2 signaling in skin, colorectal and intestinal cancers [2,4,43].

The p53 gene (*TP53*) is somatically mutated in over 50% of cancers [44]. Compared to the wild-type counterpart, p53 mutants are intrinsically more stable due to the lack of autoregulatory loops with negative regulators [45], and the stabilization of p53 mutants is a prerequisite for their gain-of-function properties [38]. We determined that the stability of p53 mutants is further amplified upon MK2 inhibition. Although we did not observe any cellular response to MK2 inhibition in serum-grown p53^R273H^ U251 cells, the treatment of serum-free stem-like p53^R110L^ JK2 cells with MK2 inhibitors caused a moderate increase in spheroid growth within 2 weeks. Thus, the administration of MK2 inhibitors to patients with mutated p53 could have dire consequences.

Our data obtained with the p53-mutated cells considerably advance the knowledge of MK2 signaling in DNA damage response. MK2 inhibition has been shown as synthetically lethal to p53 deficiency [6,7,23,24,46]. While some of these reports suggested that MK2 inhibition could be lethal to p53 mutations, none have directly manipulated MK2 in p53-mutated cells. Herein, we show that targeting MK2 in p53-mutated cells produces a phenotype that is distinct from the p53-deficient phenotype. In p53-deficient cells, MK2 inhibition led to chemo-sensitization through the accumulation of DNA damage and activation of apoptosis. However, MK2 inhibition in p53-mutated cells consistently failed to increase the efficacy of temozolomide.

This work is important in light of previous reports suggesting tissue-specific MK2 roles in inflammation [47]. In cancer, MK2 signaling was dispensable for the survival of p53^wt^ non-small-cell lung carcinoma cells [6]; however, we found that MK2 activity was necessary for the survival of p53^wt^ glioblastoma cells. In addition to p53 status and tissue-specific roles, the type of the chemotherapeutic agent is also a determinant of MK2 function in chemosensitization [24,48,49]. While MK2 inhibition improved the efficacy of doxorubicin and cisplatin [6,24], MK2 activity is crucial for the efficacy of gemcitabine [48,50]. In support of biased MK2 signaling, MK2 inhibition blocks TNF production in macrophages stimulated with lipopolysaccharide; however, when the same cells were treated with Smac mimetics, MK2 inhibition increased TNF production [9]. Taken together, the cellular response to MK2 inhibition appears remarkably flexible and depends on the cell type, the genetic background and the type of external stimuli.

Gliomas, and particularly, the most aggressive Grade IV glioblastomas, represent an unmet medical need. Our in vitro work suggests MK2 as a promising kinase for more detailed preclinical in vivo validation and clinical translation. The biomarker for the translation of MK2 inhibitors is the intact *TP53* gene. Although *TP53* mutations are the most common molecular alteration detected across all cancers [44], *TP53* mutations occur only in ~30% of glioblastomas. The dysfunctional p53 signaling in the large majority of glioblastomas is attributed to multiple oncogenic mechanisms that lead to robust p53 degradation by the proteasome [25]. Thus, ~70% of glioblastomas harbor a wild-type *TP53* gene, and the treatment of these tumors with MK2 inhibitors could lead to the activation of senescent and/or apoptotic programs. High MK2 expression in clinical samples and the correlations with short patient survival further support MK2 as a target for glioblastoma therapy. However, there is less potential for the translation of MK2 inhibitors into the therapeutic regimens of lower-grade gliomas, in which *TP53* mutations are found with a 48%–91% frequency [26].

## 4. Materials and Methods

### 4.1. Glioblastoma Cell Lines

A172 (Cat# A172), U251 (Cat# U251-MG) and U87 (Cat# U87-MG) cells were obtained from the European Collection of Cell Cultures (EACC, Salisbury, UK) through Cell Bank Australia in 2014. A172, U87 and U251 cells were authenticated by Cell Bank Australia using short tandem profiling. Identification Certificates are provided in the Appendix A. p53^R273H^ mutation in U251 was confirmed by DNA sequencing (Appendix A). FUCCI-A172 cells were generated as described [51]. All cells were cultured in DMEM medium supplemented with 10% FBS and Antibiotic–Antimycotic solution (both Life Technologies, Carlsbad, CA, USA) at 37 °C and 5% CO_2_. All cell cultures were routinely tested for mycoplasma contamination and the cumulative length of culturing did not exceed 15 passages.

### 4.2. Glioblastoma Stem-Like Cell Lines

Stem-like RN1, JK2, RKI1 and HW1 cells were derived from glioblastoma specimens [39,40]. Cells were cultured in KnockOut DMEM/F-12 basal medium supplemented with StemPro NSC SFM supplement, 2 mM GlutaMAX-ICTS, 20 ng/mL EGF, 10 ng/mL FGF-β and Antibiotic–Antimycotic solution (all Life Technologies) as adherent cells on flasks coated with MatriGel Matrix (Corning Life Sciences, Cat# 354234, Corning, NY, USA). The protocols were approved by the Human Ethics Committee of the Royal Brisbane and Women’s Hospital (RBWH 2004/161). All cell cultures were routinely tested for mycoplasma contamination and the cumulative length of culturing did not exceed 15 passages.

### 4.3. siRNA Transfection

For transient transfection, cells (1.0 × 10^5^) were transfected with MK2 *si*RNA (5 nM; Cat#4390824; 5′-CAGUAUCUGCAUUCAAUCATT-3′), *TP53 siRNA* (5 nM; Cat#4390824; 5′-GUAAUCUACUGGGACGGAATT-3′) and a scrambled sequence (5 nM, Cat#4390843) using Lipofectamine RNA iMAX (Cat# 13778150, all Life Technologies).

### 4.4. CRISPR-Cas9 Gene Editing

To generate MK2 knock-out clones, cells were serially infected with lentivirus expressing Cas9 mCherry and MK2 *sg*RNA GFP (sg#10873: 5′-TCCCAGGAGAATCTGTACGCAGGG-3′; sg#10874: 5′-TCCCGGACTGCGCCAAGGCCCGCA-3′). Constitutive Cas9 and inducible guide RNA vectors have been described [52]. Briefly, to produce MK2-*sg*RNA containing lentivirus, HEK293T cells were transfected with pFH1t-UTG-hs-MAPKAP2-Sg1s or pFH1t-UTG-hs-MAPKAP2-Sg2s, pCMV-VSVG and pCMV-RΔ8.2 using Lipofectamine 3000 (Life Technologies, Cat# L3000015). Virus-containing media, supplemented with 8 μg/mL of Polybrene (Sigma Aldrich, Cat# H9268, St. Louis, MO, USA) were added to cells. Infected cells were sorted for double positive GFP and mCherry (BD Influx Cell Sorter, Becton Dickinson, Franklin Lakes, NJ, USA), then treated with doxycycline (1 μg/mL, 72 h). Single cell clones were screened for MK2 expression by Western blotting.

### 4.5. Antibodies

Antibodies (all from Cell Signaling, Danvers, MA, USA, unless otherwise stated) against MK2 (#3042), p53 (#2524), p-p53 (#9286), p-MDM2 (#3521), lamin A/C (#2032), Hsp27 (#95357), p-Hsp27 (#2401), β-tubulin (#T8328), GAPDH (#97166), p21 (#2947), PUMA (#12450), PARP (#9532), Noxa (#14766), BAX (#5023) and anti-rabbit (#7074) and anti-mouse (#7076) HRP-linked secondary antibodies were used for immunoblotting. For immunofluorescence, antibodies against p53 (#2527), α-tubulin (#2144), γH2AX (Merck Millipore #05-636, Burlington, MA, USA), Ki67 (#9449) and anti-mouse IgG conjugated to AlexaFluor488 (Life Technologies #A10680) and anti-rabbit IgG conjugated to AlexaFluor594 (Life Technologies #A11012) were used. For immunohistochemistry, antibodies against MK2 (Abcam #ab194452, Cambridge, UK) and p-MK2 (#3007) were used.

### 4.6. Subcellular Fractionation

Cells (5.0 × 10^6^) were lysed using the NE-PER Nuclear and Cytoplasmic Extraction Reagents, as per manufacturer’s instructions (ThermoFisher Scientific, Cat# 78833, Waltham, MA, USA). Subcellular fractions were analysed by Western blotting.

### 4.7. Western Blotting

Protein concentrations were determined with Pierce BCA assay kit (ThermoFisher Scientific, Cat# 23225), following manufacturer’s instructions. Whole-cell lysates (20–40 μg) and subcellular fractions (30 μg) were resolved (1.5 h, 120 V) on 4%–12% SDS–PAGE gels and transferred onto PVDF membranes using iBlot 2, P3 for 7 min (all Life Technologies). Membranes were blocked with 5% skim milk in TBST, incubated with primary antibody in 5% BSA in TBST (overnight, 4 °C) and with secondary antibody 1% skim milk in TBST (RT, 1 h). Detection was performed with Immobilon Western HRP Substrate Luminol-Peroxidase reagent (Merck Millipore, Cat# WBKLS0500) and the ChemiDoc MP Imaging System (Bio-Rad, Hercules, CA, USA). Except for Figure 5E,F and Figure 6B,C,H; each Western blotting figure panel is composed of bands originating from the same cell lysate run on the same gel. Figure 5E,F and Figure 6B,C,H are composed of bands originating from the same cell lysate run on multiple gels, the loading controls GAPDH and β-tubulin are shown for each gel. Whole uncropped blot images for all figures containing immunoblots are provided in the Appendix A.

### 4.8. Flow Cytometry

For cell cycle analysis, cells (1.0 × 10^5^) were transfected with *si*RNA (24 h) and treated with temozolomide (72 h). Cells were fixed in 70% ice-cold ethanol (4 °C, overnight) and stained with Propidium Iodide (50 μg/mL) in the presence of RNase A (100 μg/mL, both Sigma Aldrich) for 1 h in the dark at 37 °C. Samples were analysed using FACSCalibur cytometer running CellQuest software (BD Biosciences, Franklin Lakes, NJ, USA). For the analysis of apoptosis, A172 cells (1.0 × 10^4^) were treated with MK2 inhibitor IV or *si*MK2 ± temozolomide (10 d) and stained with the Annexin V-FITC Early Apoptosis Detection kit following manufacturer’s instructions (Cell Signaling Technology, Cat# 6592). All samples were analysed within 2 h of staining on LSRFortessa X-20 cytometer running FACSDiVa v6 software (BD Biosciences) and analysis performed with FlowJo v10.3 (https://www.flowjo.com/solutions/flowjo/downloads).

### 4.9. Live Cell Imaging

FUCCI-A172 cells (8.0 × 10^3^) were treated with MK2 inhibitor IV ± temozolomide. Cells were placed in the IncuCyte S3 for 96 h (Essen Bioscience, Ann Arbor, MI, USA). Images were taken at 10× objective in green (acquisition time 300 ms) and red channels (acquisition time 400 ms). The number of cells in the G_0_/G_1_ phases (red), G_1_-S transition (yellow or overlap) and S-G2-M phases (green) were quantified using IncuCyte S3 Basic Analysis software (Essen Bioscience). The ratio of cells in each cell cycle phase were calculated as a percentage of the total number of cells counted for each treatment condition.

### 4.10. Clonogenic Assay

A172 (2.0 × 10^3^), U251 (0.5 × 10^3^), U87 (2.0 × 10^3^), RN1 (4.0 × 10^3^), HW1 (6.0 × 10^3^) and RKI1 (6.0 × 10^3^) cells were treated with MK2 inhibitor IV, PF-3644022 ± temozolomide. Cells were grown for 12 days, fixed with 50% methanol and stained with 1% Toluidine Blue (Sigma Aldrich). Colonies were counted using the ImageJ software (https://imagej.nih.gov/ij/download.html) and normalized to untreated controls (set to 100%).

### 4.11. Spheroid Assay

U87, U251 (2.0 × 10^3^) and JK2 (4.0 × 10^3^) cells were seeded onto Ultra-Low Attachment Surface coated 96-well plates (Corning) and left to form spheroids for 48 h, then treated with MK2 inhibitor IV ± temozolomide. Spheroids were imaged at Day 2 and Day 16 using the phase contrast setting on the IncuCyte S3 at 4× objective. The spheroid size was measured using the IncuCyte Zoom software (Essen Bioscience).

### 4.12. Cycloheximide-Chase Assay

Cells (3.0 × 10^5^) cells were treated with cycloheximide (30 μg/mL; 0–120 min) and cells lysed with RIPA lysis buffer. Cell lysates were analysed by Western blotting.

### 4.13. qRT-PCR

qRT-PCR was carried out according to standard protocols. Cells (3.0 × 10^5^) were treated with temozolomide (50 μM, 48 h). RNeasy mini kit (Qiagen, Hilden, Germany) was used to isolate RNA from cell lysates as per manufacturer’s instructions. cDNA was generated using Applied Biosystsems High-Capacity cDNA Reverse Transcription kit (Life Technologies, Cat# 4368814) as per manufacturer’s instructions. qRT-PCR was performed using CDKN1A (QT00062090), PML (QT00090447), YPEL3 (QT00078589), BBC3 (QT00082859), BAX (QT00031192) and PMAIP1(QT0006138) primers (all Qiagen) with KAPA SYBR FAST Universal 2X qPCR Master Mix (Kapa Biosystems, Cat# KK4602, Wilmington, MA, USA). RT-PCR was run on LightCycler 480 (Roche, Basel, Switzerland). The cycling condition were as follows: 10 min at 95 °C followed by 40 cycles, each consisting of 10 s at 95 °C and 30 s at 60 °C. Samples were run in triplicate. Threshold cycles (Ct) were calculated using the LightCycler^®^ 480 software (Roche, Basel, Switzerland). Relative quantification using the comparative Ct method was used to analyse the data output. Values were expressed as fold change over corresponding values for the control by the 2-ΔΔCt method.

### 4.14. β-Galactosidase Staining

Cells (2.0 × 10^4^) were treated with temozolomide ± MK2 inhibitor IV for 10 days, fixed and stained for β-galactosidase using the Senescence β-galactosidase Staining Kit, following manufacturer’s instructions (Cell Signaling Technology, Cat# 9860). Images were taken at 40X objective using the Zeiss Axio inverted microscope using ZEN 2 – blue edition software (Zeiss, Oberkochen, Germany) and scored using ImageJ.

### 4.15. Immunofluorescence

A172, U251 (both 3.0 × 10^4^), U87 (2.5 × 10^4^) and RN1 (5.0 × 10^4^) cells were treated with MK2 inhibitor IV ± temozolomide. Cells were fixed with ice-cold 4% PFA for 20 min at RT and blocked in 5% normal goat serum/PBS for 20 min. Cells were incubated with anti-p53 (1:50, Cell Signaling), anti-α-tubulin (1:10, Cell Signaling), γ-H2AX Ser139 (1:1000, MerckMillipore) or Ki-67 (1:400, Cell Signaling) antibodies. Secondary antibodies were Alexa488-conjugated anti-mouse IgG against p53, γ-H2AX Ser139 and Ki-67 (Life Technologies). For α-tubulin, the secondary antibody was Alexa594-conjugated anti-rabbit IgG (Life Technologies). Cell nuclei were counterstained using Prolong Gold mounting media with DAPI (Life Technologies). Images were acquired using the Zeiss Axio Scope.A1 microscope using ZEN 2 – blue edition software. Images were processed using Fiji (https://imagej.net/Fiji/Downloads).

### 4.16. Immunohistochemistry

Formalin-fixed, paraffin-embedded tissue samples from 60 patients who underwent surgery (1983–2005) at the Tampere University Hospital (Finland) were used. From these 60 patients, we obtained 60 primary tumour samples and 66 recurrent tumor samples (126 samples in total). The protocol was approved by the Ethical Review Board of Tampere University Hospital, Tampere, Finland, Dnro R07042 and National Supervisory Authority for Welfare and Health, Finland, Dnro V/78697/2017. Immunohistochemistry was performed on a Bond III autostainer (Leica Microsystems, Weltzar, Germany). Microarray sections (5 μm; 0.28 mm^2^) were deparaffinized and rehydrated, followed by quenching and heat-induced epitope retrieval. Sections were incubated with MK2 and p-MK2 antibodies and visualized using the Bond Polymer Refine Detection kit (Leica Microsystems) and 3,3′-diaminobenzidine according to manufacturer’s recommendations. The tissue sections were counterstained with haematoxylin. MK2 and p-MK2 immunoreactivity was evaluated by a neuropathologist using a Zeiss Axio microscope. Immunoreactivity was scored negative (<5% positive cells) or positive (>5% positive cells). The positive samples were assigned scores 1+ (weak), 2+ (moderate) or 3+ (strong) based on the highest staining intensity observed in the sample.

### 4.17. Statistics

Kaplan–Meier survival plots were evaluated by log-rank analysis. Significance for tissue microarrays was determined using chi-square test, except for the age correlation which was performed with the Mann–Whitney test. All immunoblots and immunofluorescence images are representative of at least three independent experiments, exact numbers of independent repeats are indicated in the figure legends. All functional assays were repeated at least three times with minimum two replicates, exact numbers of independent repeats are indicated in the figure legends. All bar graph data represent mean ± SEM. For clonogenic survival, treatments were normalized to untreated control set as 100%. Growth of spheroids was expressed as fold change calculated from size at Day 16 over size at Day 2. For β-galactosidase staining, % of positive cells were calculated from the number of blue cells over the total number of cells counted. For cycloheximide-chase experiments, bands were normalized to GAPDH and protein half-life calculated by non-linear regression analysis. For qRT-PCR, all values were normalized to GAPDH and fold change calculated using 2-ΔΔCt method compared to parental control. Significance was determined with *t*-test or ANOVA (indicated in figure legends) using GraphPad Prism.

## 5. Conclusions

By elucidating the MK2-mediated control mechanism of p53 degradation in glioblastoma cells, this study proposes MK2 as a potential drug target for ~70% of glioblastomas carrying the intact *TP53* gene. For the first time, our findings in p53-mutated cells reveal that the administration of the MK2 inhibitor may trigger the undesirable stabilization of p53 mutants, thereby causing the acceleration of the malignant processes. These findings are highly relevant since p53 mutations occur in over 50% of all cancers [44].

## Figures and Tables

**Figure 1 cancers-12-00654-f001:**
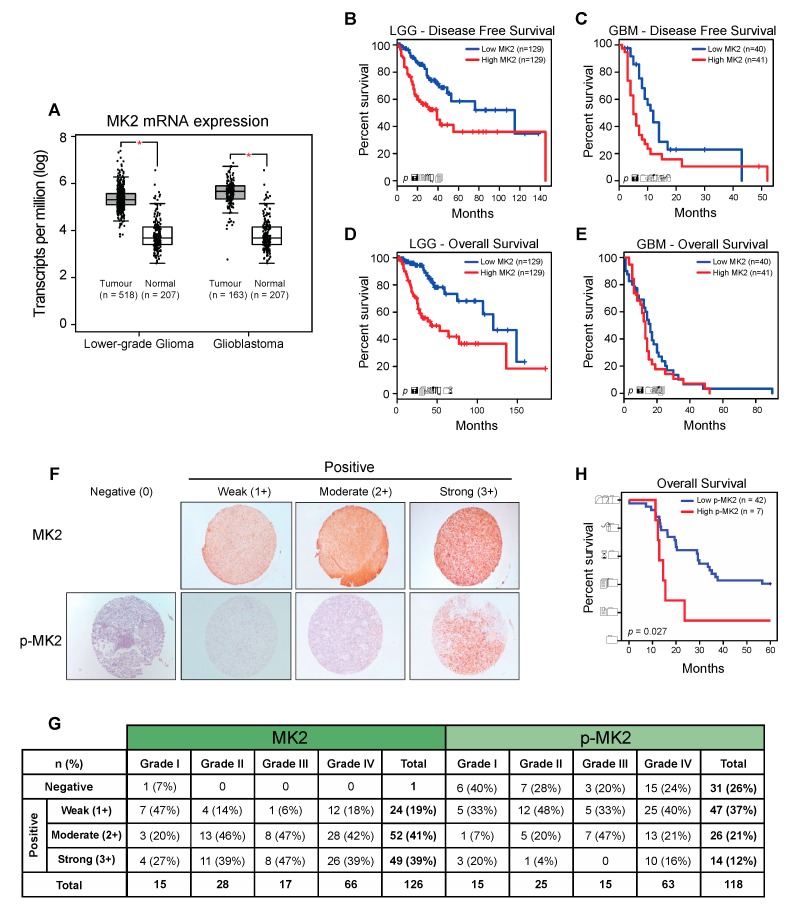
MK2 activity in gliomas correlates with poor prognosis. (**A**) MK2 mRNA expression in lower-grade glioma and glioblastoma compared to normal brain tissue. Figure was generated by GEPIA (mean ± SD, ANOVA, * *p*  <  0.05). (**B**–**E**) Log-rank survival analysis of lower-grade glioma (LGG) and glioblastoma (GBM) patients based on the MK2 mRNA expression. Figure was generated by GEPIA (low MK2: bottom 25%; high MK2: top 25%). (**F**,**G**) Representative images and summary of MK2 and p-MK2 immunoreactivity in glioma tissue microarrays. (**H**) Log-rank survival analysis of lower-grade glioma and glioblastoma patients based on the p-MK2 expression (low p-MK2: 0, 1+, 2+ scores; high p-MK2: 3+ score).

**Figure 2 cancers-12-00654-f002:**
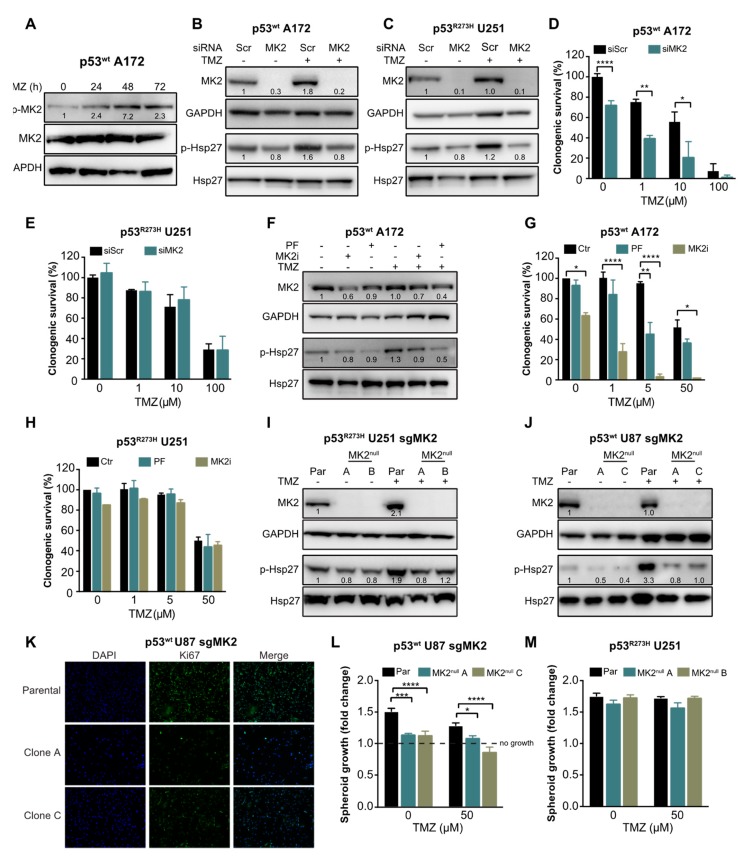
MK2 inhibition attenuates proliferation of p53^wt^ glioblastoma cells. (**A**) Cells were treated with temozolomide (50 µM) for 0–72 hr and lysates immunoblotted against indicated antibodies. (**B,C**) Scramble (Scr) or MK2 siRNA transfected cells were treated with temozolomide (TMZ, 50 µM, 48 hr) and lysates immunoblotted against indicated antibodies. (**D**,**E**) Scramble or MK2 siRNA transfected cells were treated with temozolomide and colonies quantified after 12 days of treatment. (**F**) Cells were treated with PF-3644022 (PF, 3 µM), MK2 inhibitor IV (MK2i, 1.5 µM) and temozolomide (50 µM, 48 hr) and lysates immunoblotted against indicated antibodies. (**G**,**H**) Cells were treated with PF-3644022 (3 µM), MK2i (1.5 µM) and temozolomide and colonies quantified after 12 days of treatment. (**I**,**J**) Parental (Par) and MK2^null^ cells were treated with temozolomide (50 µM, 48 hr) and lysates immunoblotted against indicated antibodies. (**K**) Parental and MK2^null^ cells were stained with Ki67 antibody and DAPI. (**L**,**M**) Parental and MK2^null^ spheroids were treated with temozolomide and growth calculated as the spheroid size on Day 16 over Day 2 (treatment day). All immunoblot images and band intensities are representatives of 3-4 independent experiments. Bar graphs for clonogenic and spheroid growth assays represent mean ± SEM from 3 independent experiments performed in duplicate (* *p* < 0.05; ** *p* < 0.01; *** *p* < 0.001; **** *p* < 0.0001; one-way ANOVA followed by Bonferroni’s post-test for multiple comparison). Immunofluorescence images are representatives of 3 independent experiments.

**Figure 3 cancers-12-00654-f003:**
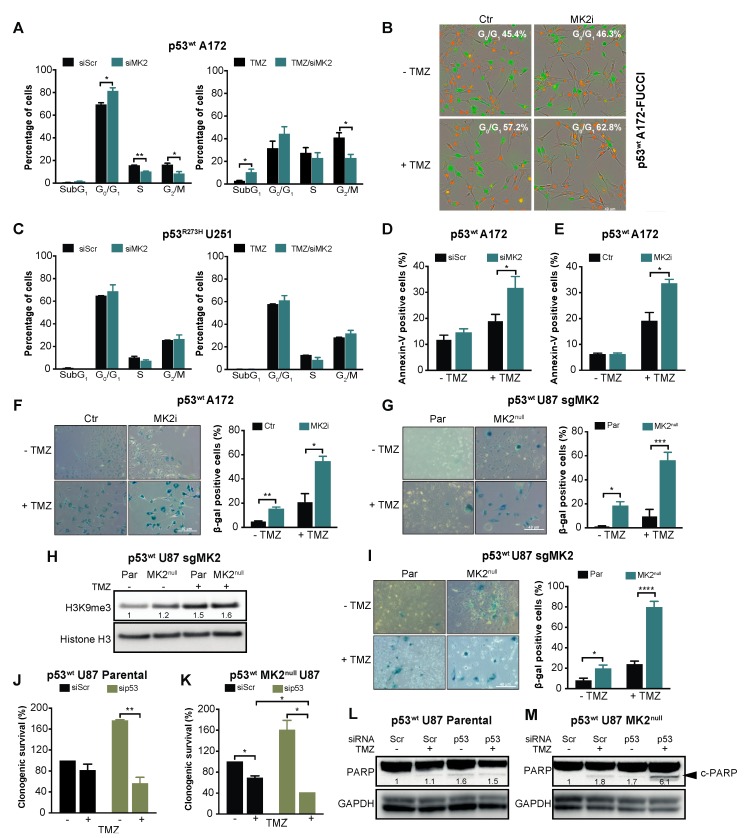
MK2 inhibition induces senescence and apoptosis in p53^wt^ glioblastoma cells. (**A**) Scramble (Scr) or MK2 siRNA transfected cells were treated with temozolomide (TMZ, 50 µM, 72 hr) and PI staining analyzed by flow cytometry. (**B**) Cells were treated with MK2 inhibitor IV (MK2i, 1.5 µM) and temozolomide (10 µM, 96 hr) and quantification of G_0_/G_1_ (red cells) was performed with IncuCyte S3. (**C**) Scramble or MK2 siRNA transfected cells were treated with temozolomide (50 µM, 72 hr) and PI staining analyzed by flow cytometry**.** (**D**) Scramble or MK2 siRNA transfected cells were treated with temozolomide (25 µM, 10 days) and Annexin-V staining analyzed by flow cytometry. (**E**) Cells were treated with MK2 inhibitor IV (1.5 µM) and temozolomide (25 µM, 10 days) and Annexin-V staining analyzed by flow cytometry. (**F**) A172 cells were treated with MK2 inhibitor IV (1.5 µM) and temozolomide (25 µM, 10 days) and stained for β-galactosidase (β-gal). (**G**) U87 Parental (Par) and MK2^null^ (clone C) cells were treated with temozolomide (25 µM, 10 days) and stained for β-galactosidase. (**H**) Parental and MK2^null^ (clone C) cells were treated with temozolomide (25 µM, 10 days) and lysates immunoblotted against indicated antibodies. (**I**) Parental and MK2^null^ (clone C) cells were treated with temozolomide (25 µM, 20 days) and stained for β-galactosidase. (**J**,**K**) Parental and MK2^null^ (clone C) cells were transfected with scramble or p53 siRNA, treated with temozolomide (25 µM) and colonies quantified after 12 days of treatment. (**L**,**M**) Parental and MK2^null^ (clone C) cells were transfected with scramble or p53 siRNA, treated with temozolomide (25 µM, 7 days) and lysates immunoblotted against indicated antibodies. All immunoblot including band intensities and microscopic images are representatives of 3 independent experiments. Bar graphs represent mean ± SEM from 3 independent experiments (* *p* < 0.05; ** *p* < 0.01; *** *p* < 0.001; **** *p* < 0.0001; paired *t*-test).

**Figure 4 cancers-12-00654-f004:**
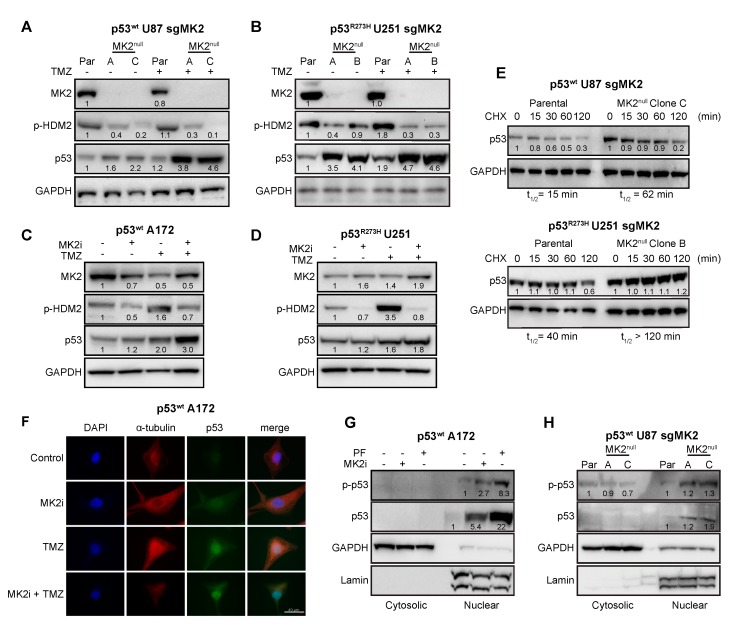
FMK2 inhibition increases p53 expression. (**A,B**) Parental (Par) cells and MK2^null^ clones were treated with temozolomide (TMZ, 50 µM, 48 hr) and lysates immunoblotted against indicated antibodies. (**C,D**) Cells were treated with MK2 inhibitor IV (MK2i, 1.5 µM) and temozolomide (TMZ, 50 µM, 48 hr) and lysates immunoblotted against indicated antibodies. (**E**) Parental cells and MK2^null^ clones were treated with cycloheximide (CHX, 2 µg/mL) and lysates immunoblotted against indicated antibodies. The p53 half-life (t_1/2_) was calculated by nonlinear regression analysis of immunoblotting signal intensity (BioRad Image Lab). (**F**) Cells were treated with MK2 inhibitor IV (MK2i, 1.5 µM) and temozolomide (50 µM, 48 hr) and fixed cells stained with indicated antibodies and DAPI. (**G**) Cells were treated with PF-3644022 (PF, 3 µM) or MK2 inhibitor IV (MK2i, 1.5 µM, 24 hr) and sub-cellular fractions immunoblotted against indicated antibodies. (**H**) Cytosolic and nuclear extracts of parental cells and MK2^null^ clones were immunoblotted against indicated antibodies. All immunoblots including band intensities and immunofluorescence images are representatives of 3-4 independent experiments.

**Figure 5 cancers-12-00654-f005:**
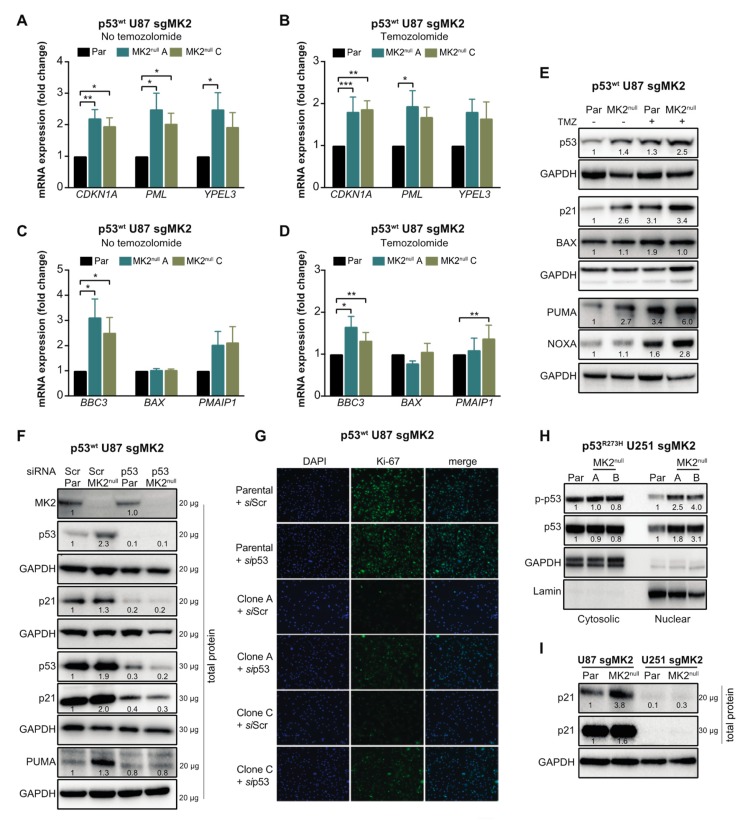
p53^wt^ silencing rescues the anti-proliferative efficacy of MK2 inhibition. (**A**–**D**) Parental (Par) cells and MK2^null^ clones were treated with temozolomide (TMZ, 50 µM, 48 hr) and nuclear extracts analyzed by qRT-PCR for senescence (**A**,**B**) and apoptosis (**C**,**D**) genes. Bar graphs represent mean ± SEM from 3 independent experiments (* *p* < 0.05; ** *p* < 0.01; *** *p* < 0.001; paired *t*-test). (**E**) Parental and MK2^null^ (clone A) cells were treated with temozolomide (50 µM, 48 hr) and lysates immunoblotted against indicated antibodies**.** (**F**) Parental and MK2^null^ (clone A) cells were transfected with scramble (Scr) or p53 siRNA for 72 hr, and lysates immunoblotted against indicated antibodies. (**G**) Parental and MK2^null^ (clones A and C) cells were transfected with scramble or p53 siRNA for 72 hr. Fixed cells were stained with Ki67 antibody and DAPI. (**H**) Cytosolic and nuclear extracts of parental and MK2^null^ (clones A and B) cells were immunoblotted against indicated antibodies. (**I**) Lysates of parental and MK2^null^ (clone A for U87; clone A for U251) cells were immunoblotted against indicated antibodies. All immunoblots including band intensities and immunofluorescence images are representatives of 3 independent experiments.

**Figure 6 cancers-12-00654-f006:**
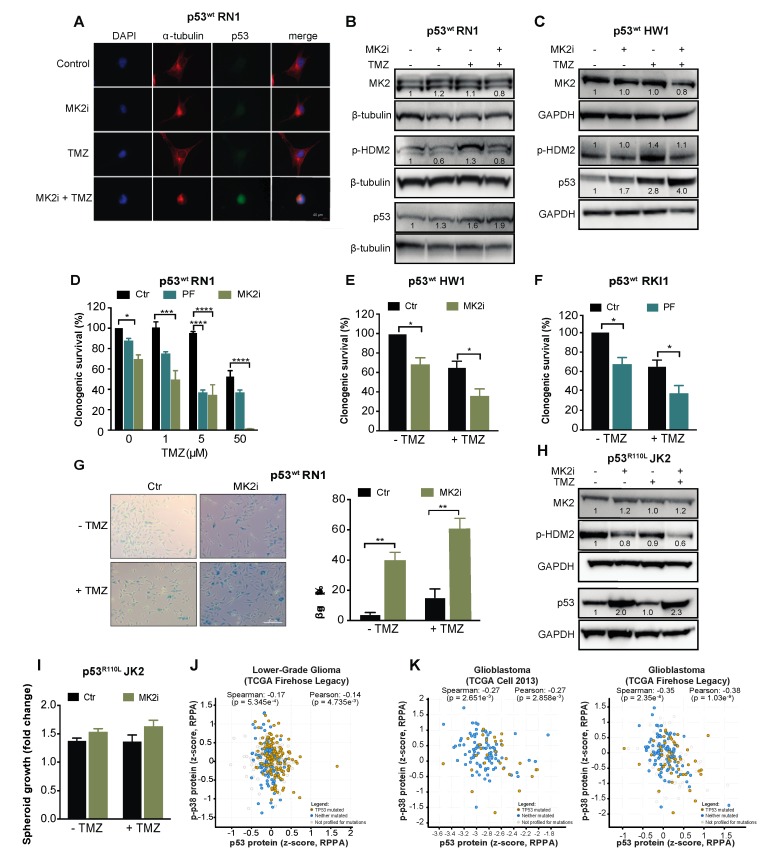
MK2 inhibitors attenuate proliferation of p53^wt^ glioblastoma stem-like cells. (**A**) Cells were treated with MK2 inhibitor IV (MK2i, 1.5 µM) and temozolomide (TMZ, 50 µM) for 48 hr and fixed cells stained with indicated antibodies and DAPI. (**B**,**C**) Cells were treated with MK2i (1.5 µM) and temozolomide (50 µM) for 48 h and lysates immunoblotted against indicated antibodies. (**D**–**F**) Cells were treated with MK2 inhibitor IV (1.5 µM), PF-3644022 (PF, 3 µM) and temozolomide for 12 days and colonies quantified using ImageJ. (**G**) Cells were treated with MK2 inhibitor IV (1.5 µM) and temozolomide (25 µM) for 10 days and stained for β-galactosidase (β-gal). (**H**) JK2 cells were treated with MK2 inhibitor IV (1.5 µM) and temozolomide (50 µM) for 48 h and lysates immunoblotted against indicated antibodies. (**I**) JK2 spheroids were treated with temozolomide and growth calculated as the spheroid size on Day 16 over Day 2 (treatment day). (**J**,**K**) Correlations of p-p38 MAPK and p53 protein expression in the TCGA cohorts of lower-grade glioma (n = 435) and glioblastoma (n = 214 for Cell 2013, n = 244 for Firehose Legacy; cbioportal.org). All immunoblot, immunofluorescence and microscopic images are representatives of 3 independent experiments. Bar graphs for clonogenic and spheroid growth assays are mean ± SEM from 3 independent experiments performed in duplicate (* *p* < 0.05; ** *p* < 0.01; **** *p* < 0.0001; one-way ANOVA followed by Bonferroni’s posttest for multiple comparison in D and paired *t*-test in E,F,G).

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
