# Peer review of "MK2 Inhibition Induces p53-Dependent Senescence in Glioblastoma Cells"

_cancers, 2020, doi:10.3390/cancers12030654_

Round 1
Reviewer 1 Report
To the authors:
In the manuscript by Phoa et al., the authors analyzed the possibility to target glioblastoma via inhibition of MAPK-activated protein kinase 2 (MK2) in the context of mutated and wildtype p53. Using this approach, the authors present in interesting strategy for a cancer that, despite decades of research, is still largely uncurable. Furthermore do the authors nicely illustrate that the genetic makeup (p53 mutant vs wildtype) has a huge impact on the outcome of this therapy and might in fact be useful for patient stratification. However, I am wondering that given the high and widely known genetic heterogeneity of GBM if truly all tumor cells will have the same p53 phenotype and/or if MK2-inhibition rather drives a p53 mutant clonal expansion? Maybe the authors can comment on that. Nonetheless was this study nicely done, very well written and presents the key questions with a great number of different experiments including patient data, conventional cell lines and spheroids.
Prior publication some points need to be addressed
- Are the cell lines authenticated? If yes, please provide the authentification data; if not, please authenticate.
- Figure 1 B-E and H: There are some symbol in the figure that appear to not belong there. Please carefully check if those belong there. If so, please explain and increase readability; if not: please remove.
- Line 86-87 and Fig. 1: How do you obtain 126 samples from 50 patients? Do the samples include recurrent and/or progressing cancers? Please explain in more detail. If some samples are derived from different timepoints of the same patients: how does this influence the expression and/or activation status?
- Lines: 286-287: The conclusion that MK2 inhibition for the therapy of GBM is a little far-fetched based on the existing. Promising as these data may be the authors only present in vitro data, hence the data suggest that MK2 is a suitable target for more directed preclinical testing and the authors should re-phrase their discussion and conclusion accordingly.
Minor points:
- Line 44-46: please place the reference at the point of reference (e.g.: “… skin tumors” [Ref X]
- Line 47: please state some examples and refer to those instead of listing Ref 5-10 without further comment
- Line 49: please provide the “limited data”
- Line 60: TMZ is an alkylating agent.
- Line 298: The heading is somethat misleading. Please divide this into subsections (e.g. cell lines and compounds; viral transduction etc.). This will guide potential readers better to your method section.
- I am unsure whether this was done by authors or if this happened during processing at Cancers, but to place Fig. 5 and 6 into the materials and methods sections is quite misleading and might cause some confusion. Hence I would wish for these figures to be placed in the result section.
Thank you and Good luck!
Author Response
We thank Reviewer 1 for very helpful feedback.
I am wondering that given the high and widely known genetic heterogeneity of GBM if truly all tumor cells will have the same p53 phenotype and/or if MK2-inhibition rather drives a p53 mutant clonal expansion? Maybe the authors can comment on that. We agree. If a tumour contains a small subpopulation of p53mutated cells, then MK2 inhibition would most likely drive expansion of these p53mutant clones.
- Are the cell lines authenticated? If yes, please provide the authentification data; if not, please authenticate. A172, U87 and U251 cells were authenticated by Cell Bank Australia using short tandem profiling. Identification Certificates are provided in the Supplementary Material. p53 R273H mutation in U251 was confirmed via DNA sequencing (Supplementary Figure S9). This is now included in Materials and Methods (page 7, lines 306 – 308)
- Figure 1 B-E and H: There are some symbol in the figure that appear to not belong there. Please carefully check if those belong there. If so, please explain and increase readability; if not: please remove. We believe this has occurred through merging of the files. The survival curves in Figure 1B-E and 1H contain colour scheme (blue for low MK2/pMK2; red for high MK2/pMK2) and the p value.
- Line 86-87 and Fig. 1: How do you obtain 126 samples from 50 patients? Do the samples include recurrent and/or progressing cancers? Please explain in more detail. If some samples are derived from different timepoints of the same patients: how does this influence the expression and/or activation status? There were 60 glioma patients in this study. We apologize for the typo error with the patients number. This has been corrected and clarified through the manuscript. From these 60 patients we obtained 60 primary tumour samples and 66 recurrent tumor samples (126 in total). Thus, several samples were obtained from the same patient, however at different timepoints. Increased MK2 expression was associated to recurrent glioblastomas (p = 0.009; chi-square test) and p-MK2 expression correlated with recurrent glioblastomas (p = 0.049, chi-square test), as mentioned in results and Supplementary Figure S1C.
- Lines: 286-287: The conclusion that MK2 inhibition for the therapy of GBM is a little far-fetched based on the existing. Promising as these data may be the authors only present in vitro data, hence the data suggest that MK2 is a suitable target for more directed preclinical testing and the authors should re-phrase their discussion and conclusion accordingly. We have re-worded this section in the Discussion to “Our in vitro work suggests MK2 as a promising kinase for more detailed preclinical in vivo validation and clinical translation” (page 6, lines 291-292) and in the Conclusion to “this study proposes MK2 as a potential drug target for ~70% of glioblastomas” (page 10, line 455).
Minor points:
- Line 44-46: please place the reference at the point of reference (e.g.: “… skin tumors” [Ref X]. We have added citation at the point of reference as suggested (page 1; lines 44-46)
- Line 47: please state some examples and refer to those instead of listing Ref 5-10 without further comment. We have added examples of cancer drugs that were successfully used in combination with MK2 inhibition and added citation at the point of reference (page 2; lines 46-48)
- Line 49: please provide the “limited data”. We have included references [10-14], which to our best knowledge are the only publications investigating MK2 in glioma context. We have expanded on references relevant to our current paper in more detail in the next paragraph (page 2, lines 57-60) as we feel that introducing glioma tumour grades and the need for new glioma targets (page 2, line 51-56) should be discussed before MK2-glioma context (page 2, line 57-60) .
- Line 60: TMZ is an alkylating agent. This has been amended (page 2, line 63).
- Line 298: The heading is somethat misleading. Please divide this into subsections (e.g. cell lines and compounds; viral transduction etc.). This will guide potential readers better to your method section. This has been amended (page 7), new subsections have been introduced for clarity.
- I am unsure whether this was done by authors or if this happened during processing at Cancers, but to place Fig. 5 and 6 into the materials and methods sections is quite misleading and might cause some confusion. Hence I would wish for these figures to be placed in the result section. This has occurred during the processing since we have provided all Figured at the end of the document. In the Revision, we will also provide Figures at the end of the manuscript.
Reviewer 2 Report
In this interesting work, the authors studied the roles of MAPK-activated protein kinase 2 (MK2) in malignant glioma cell lines and patient-derived stem cells using genetic and pharmacological approaches. They show that the expression/activity of MK2 correlates with poor glioma prognosis. MK2 inhibition increases the expression of P53 and attenuates the growth of glioblastoma cells expressing normal P53 but not deficient P53. Finally, the anti-proliferative effect of MK2 inhibition appears to be mediated by P53 in glioblastoma cells. The study overall was well designed and performed, and the manuscript is also well written. I just have a few commons for the authors’ consideration:
- The glioma cell lines used in the study should be authenticated.
- The TCGA and GTEx data analysis needs to be stratified by IDH mutation status.
- How was the p53R273H U251 cell line generated and validated?
- Wild type A172 and U87 were used in this study, but I think p53wt U251 should also be used and compared with p53R273H U251.
Author Response
We thank Reviewer 2 for very helpful feedback.
1. The glioma cell lines used in the study should be authenticated. A172, U87 and U251 cells were authenticated by Cell Bank Australia using short tandem profiling. Identification Certificates are provided in the Supplementary Material. p53 R273H mutation in U251 was confirmed via DNA sequencing (Supplementary Figure S9). This is now included in Materials and Methods (page 7, lines 306 – 308)
2. The TCGA and GTEx data analysis needs to be stratified by IDH mutation status. GEPIA that uses TCGA and GTEx data provides differential expression analysis, profiling plotting, correlation analysis, patient survival analysis, similar gene detection and dimensionality reduction analysis. However, it does not include an option to analyse data on gene mutations. We used GEPIA for correlation of gene expression with survival and unfortunately cannot do stratification based on IDH mutation.
3. How was the p53R273H U251 cell line generated and validated? U251 cell line naturally carries the p53 R273H mutation, which we confirmed by DNA sequencing (Supplementary Figure S9). This cell line was not generated but purchased (in 2014) and authenticated (in 2020) from the European Collection of Cell Cultures (EACC, Salisbury, UK) through Cell Bank Australia. Certificates are provided in the Supplementary Material.
4. Wild type A172 and U87 were used in this study, but I think p53wt U251 should also be used and compared with p53R273H U251. We agree, however as mentioned above U251 naturally carry p53 R273H. To generate p53wt U251 cells, we would have to first perform CRISPR-Cas9 knock-out of p53 R273H and then reconstitute cells with p53wt constructs. However, this is not trivial as we cannot use conventional CRISPR technique to reconstitute as the p53wt would get cleaved by Cas9. Finding an alternative approach would be a very time-consuming challenge and we believe that our data in several p53wt cells (U87, A172, RN1, HW1, RKI1) are supporting our conclusions.